# Back Pain and Schoolbags among Adolescents in Abha City, Southwestern Saudi Arabia

**DOI:** 10.3390/ijerph17010005

**Published:** 2019-12-18

**Authors:** Abdullah Assiri, Ahmed A. Mahfouz, Nabil J. Awadalla, Ahmed Y. Abolyazid, Medhat Shalaby

**Affiliations:** 1Department of Internal Medicine, College of Medicine, King Khalid University, Abha 61421, Saudi Arabia; aassiri@yahoo.com (A.A.); medhatshalaby57@gmail.com (M.S.); 2Department of Family and Community Medicine, College of Medicine, King Khalid University, Abha 61421, Saudi Arabia; njgirgis@yahoo.co.uk (N.J.A.); drzizous2000@yahoo.com (A.Y.A.); 3Department of Epidemiology, High Institute of Public Health, Alexandria University, Alexandria 21511, Egypt; 4Department of Community Medicine, College of Medicine Mansoura University, Dakahlia Governorate 35516, Egypt; 5Department of Rheumatology, Faculty of Medicine, Al Azhar University, Cairo 11651, Egypt

**Keywords:** back pain, adolescents, schoolbags, Saudi Arabia

## Abstract

The incorrect carriage of schoolbags and heavy bags may result in back pain (BP) in adolescents. Our objective was to assess the problem of BP and associated factors among adolescents. This was a cross-sectional study targeting intermediate and secondary school adolescents in Abha City, Saudi Arabia. An anonymous questionnaire for BP was used. Student body weight and the weight of their full schoolbag were measured. The study included 876 adolescents. An overall prevalence of BP of 39.4% (95% CI: 35.3–43.7) was found. Upper BP (UBP), middle BP (MBP), and lower BP (LBP) amounted to 14.5%, 13.4%, and 11.5%, respectively. Associated significant factors with MBP were carrying the bag on one side compared to on the back (adjusted odds ratio (aOR) = 2.13, 95% CI: 1.20–3.73) and being at intermediate level compared to secondary (aOR = 1.56, 95% CI: 1.04–2.40). On the other hand, gender and schoolbag weight/body weight percent were not found to be significantly associated with MBP. None of the studied factors were significantly associated with overall BP, UBP, or LBP. BP is a prevalent among adolescents in southwest Saudi Arabia. One-sided schoolbag carriage is a significant associated factor for middle back pain. Parents and teachers should encourage students to consider the correct way to carry schoolbag.

## 1. Introduction

Carrying schoolbags and attending school for a considerable period of time is a scheduled daily activity for adolescents. Recently, the relationship between carrying schoolbags and back pain (BP) has received considerable attention. The incorrect carriage of schoolbags, and carrying heavy bags, may result in back pain in adolescents [1,2,3].

It is recommended that the weight of the full schoolbag should not exceed 10%–15% of body weight [4]. This is justified by the physiological and biomechanical impacts of carrying heavy schoolbags [4]. However, this limit is often exceeded. Reports from different regions in the world reveal that a considerable proportion of school children carry schoolbags exceeding 20% of their body weight [5,6,7].

Moreover, previous studies have shown that the schoolbag carriage method is also related to back pain. A bag carried on one side is associated with more back pain than wearing a two-shouldered bag [8,9].

Studies have reported that the prevalence of back pain in school children ranges from 25% to 55% in those aged between 10 and 15 years [1,10,11]. Back pain in adolescents is an important issue because it increases the risk of having chronic back pain in adulthood [12].

In Saudi Arabia, particularly the southwest region, the extent of back pain problems and their relationship with schoolbag weight and carriage method among school adolescents has not been fully settled. Therefore, the objectives of the present study were to assess the problem of back pain among school adolescents, explore schoolbag weight as a percentage of body weight, find a new optimal cutoff point that indicates BP, identify the methods of carrying schoolbags, and to correlate back pain with these factors.

## 2. Materials and Methods

### 2.1. Design

This was a cross-sectional study.

### 2.2. Target Population

The target population was intermediate and secondary school age adolescents in Abha City, Aseer Region, Southwestern Saudi Arabia. The intermediate and secondary levels in Saudi Arabia include students in grades 7 to 9 and 10 to 12, respectively. The students’ ages range from 13–15 years at the intermediate level and 16–18 years at the secondary level.

### 2.3. Sample Size Determination and Sampling Technique

Using the WHO Manual for Sample Size Determination in Health Studies [13], with a conservative anticipated proportion of back pain of 52.2% among school children in Riyadh [14] and absolute precision of 2% at the 95% confidence interval, the minimum sample size required for the study was calculated to be 600 adolescents. A multistage stratified proportional cluster random sample method was followed to select the study population. The stratification factors taken into consideration were the relative number of students in intermediate and secondary schools, male/female differences, and the district in which they lived. The students were recruited from 13 intermediate and secondary schools in Abha city. Within each school, one class was selected randomly from each of the three grades (7–9 and 10–12).

### 2.4. Questionnaire Interview

An anonymous paper-based questionnaire was distributed to each student in their own classroom. The questionnaire included personal and demographic data, and preferred method of schoolbag carriage (any side or always on the back). Back pain (BP) was assessed using the following question: “During the past three months have you felt any pain or ache in the back area for a long time at school, which has lasted for one day or longer?” The response options were “yes”, “no”, or “I don’t know”. The last option was regarded as “no”. Students positive to back pain were asked to indicate the exact site of the pain (upper, middle, or lower).

### 2.5. Adolescent Weight and Schoolbag Weight Measurements

The adolescent’s weight was taken while the student was standing erect without shoes and with minimal clothing. The weight of the student’s full schoolbag was measured in kilograms. Bag weight/body weight percent was computed. All measurements were taken by trained examiners.

### 2.6. Funding and Ethical Approval

The research proposal was revised and accepted by the ethical committee of King Khalid University (received 9 March 2014). Written consent was obtained from the students’ parents. Students omitted from the study were those absent on the day of the visit or those with a parental request to not participate in the study.

### 2.7. Data Analysis

Data were analyzed using SPSS. Chi square test was used as a test of significance at 5% level. Prevalence rates were calculated with concomitant 95% confidence intervals (95% CIs).

A receiver operating characteristic (ROC) curve was constructed to examine the predictive performance of schoolbag load as a percent of body weight in identifying children with back pain (overall, upper, middle, and lower). The graphical plot demonstrated the performance of the cutoff points in terms of sensitivity versus 1-specificity. The area under the curve (AUC) is a measure of the accuracy of a test or cutoff point. The AUC value lies between 0.5 and 1, where less than 0.6 denotes a poor classifier and 1 denotes an excellent classifier.

To study the factors associated with BP, multivariable binary logistic regression analysis was used. Adjusted odds ratio (aOR) and the concomitant 95% CI were computed. Variables included in the model were gender, level (intermediate or secondary), schoolbag carriage method (on the back or on one side) and bag weight/body weight percent.

## 3. Results

### 3.1. Description of the Study Sample

The study included 876 adolescents who were students in intermediate (439, 50.1%) and secondary schools (437, 49.9%). The sample comprised 471 males (53.8%) and 405 females (46.2%). Their age ranged from 13 to 19 years with a mean of 15.88 ± 1.76 years and a median of 16 years. Regarding fathers’ education, 39.5% (346) were university educated and 27.9% were educated at the secondary level (244). Regarding mothers’ education, the majority were educated to less than secondary level (49.4%, 433) and 23.2% were secondary level educated (203).

### 3.2. Prevalence of BP

This study revealed that 345 students reported having BP, giving an overall prevalence of 39.4% (95% CI: 35.3–43.7). Among those having BP, 182 (47.2%) reported having pain at school and 149 (43.2%) reported increase in pain when carrying heavy objects. Only 31 students (9.0%) asked for medical consultations.

Table 1 shows that the prevalence of upper back pain (UBP) amounted to 14.5% (95% CI: 12.1–17.2). Similarly, the prevalence of middle back pain (MBP) amounted to 13.4% (95% CI: 11.1–15.5), and the prevalence of lower back pain (LBP) was 11.5% (95% CI: 9.4–13.9).

### 3.3. Prediction of BP by Schoolbag Weight/Student Body Weight Percent

Table 2 shows ROC curve analysis of the discrimination of schoolbag weight/body weight percent for back pain. The ability of schoolbag weight/body weight percent, expressed as a continuous variable, to discriminate between those with and without overall, upper, middle, and lower back pain was poor (AUC < 0.6). The corresponding AUC values were 0.501, 0.570, 0.521, and 0.564, respectively. Based on these results, ROC analysis failed to identify new optimal cutoff points.

### 3.4. Characteristics of the Students’ Schoolbags

Table 3 shows the characteristics of students’ schoolbags. The weight of a full schoolbag ranged from 1 to 10.5 kg with an average of 4.97 ± 1.60 kg and a median of 4.9 kg. The average weight of the bags was not statistically different (*p* = 0.623) among intermediate students (4.95 ± 1.41) compared to secondary students (5.00 ± 1.77). Schoolbag weight/body weight percent of the study sample ranged from 3.22% to 16.8% with an average of 9.14% ± 4.9% and a median of 8.6%. Regarding the distribution of schoolbag weight/body weight percent by level, the frequency of students with a value of ≥10% was higher among intermediate- (42.6%) than secondary-level students (31.6%). The distribution was statistically significant (*p* = 0.001). This cutoff point was based on the literature. Regarding schoolbag carriage methods, most of the intermediate students carried their bag on their backs (254, 57.9%), compared to 21.7% (95) in secondary-level students. The difference was statistically significant (*p* = 0.001).

### 3.5. Factors Associated with BP

Table 4 shows multivariable analysis of potential factors associated with back pain among the study sample of school adolescents. Regarding MBP, intermediate-school adolescents had significantly higher probability of having pain (aOR = 1.56, 95% CI: 1.04–2.40) compared to secondary school adolescents. Similarly, students carrying bags on one side had significantly higher probability of having MBP (aOR = 2.13, 95% CI: 1.20–3.73) compared to students who carried bags on their back. On the other hand, gender and schoolbag weight/body weight percent were not found to be significantly associated with MBP.

The study showed that none of the studied factors were found to be significantly associated with overall BP, UBP, or LBP.

## 4. Discussion

Emerging data suggest that back pain in adolescents is responsible for a substantial disability burden and consumes considerable healthcare services. Of further concern is the fact that back pain during this period of life may have health implications in adulthood. Although the enormous disability burden of back pain in adults is well documented, the consequences of the condition in children are not so well acknowledged [15]. The attitude toward back pain in children and adolescents has undergone a large change in the past 15–20 years. Report of back pain in childhood was previously considered rare, and a sign of serious underlying pathology. In fact, several clinical practice guidelines include “age under 20 years” as a red flag for back pain assessment [16]. However, more recent studies have indicated that the condition is common, and it is usually not possible to diagnose a specific patho-anatomical cause for the pain [17].

The present study reported a prevalence rate of back pain (BP) of 39.4% among adolescents. Figures for UBP, MBP, and LBP were 14.5%, 13.4%, and 11.5%, respectively. A systematic review reported wide ranges of prevalence estimates of back pain due to differences in study population, definition of back pain, study design, and prevalence period. Point prevalence estimates ranged from 3% to 39% [15]. A recent study among Nigerian adolescents reported an overall BP prevalence of 17%. The corresponding figures for UBP, MBP, and LBP were 19%, 15%, and 25%, respectively [18]. In Tunisia, a figure of 32% of BP was reported among secondary school students [19]. Similarly, a figure of 32.1% was found in a recent study in Iran [20]. In Sweden, a figure of 44.7% was reported [21]. In Spain, a figure of 39.8% for LBP was reported [22]. In Saudi Arabia, a recent study reported a figure of 31.9% for LBP among adolescents [23].

A recent systematic review revised important associated factors related to BP among adolescents. The identified associated factors were gender, age, weight, posture, and schoolbag [15]. The present study addressed the following associated factors: gender, level (as a proxy for age), and schoolbag.

The present study showed that gender was not significantly associated with back pain among adolescents. A study in Australia showed that adolescent girls appear to be at higher risk of reporting back pain than boys [24]. Another study in Nigeria showed that gender was not associated with BP among secondary-school adolescents [25]. The role of gender in BP showed inconsistency in systematic reviews [15].

The results of the present study revealed that school students were mostly used to carrying schoolbags on one side (60.2%). However, this finding is not in agreement with the results of other studies [1,6,26,27], which reported that most students carried their schoolbags over two shoulders. Moreover, this finding is not in accordance with the guidelines for the safe carriage of schoolbags. Most organizations’ guidelines encourage the use of both backpack straps to carry the schoolbag [28,29]. The results of the present study reveal a significant relationship between the schoolbag carriage method and the prevalence of middle back pain (MBP). The risk of MBP was significantly higher among students who carried their schoolbag on one side. This finding is in agreement with the results of other studies [1,5,30,31,32]. It seems that carrying the schoolbag on two shoulders helps to maintain a correct posture and reduce the risk of back pain among students. With regards to the one-side carriage method, studies have indicated that students who carry their schoolbag on one side foster lateral curvature of the spine, impaired posture, and consequently, increased probability of musculoskeletal pain [31,33].

In the present study, the average schoolbag weight was 4.97 ± 1.60 kg and the average relative weight was 9.14% ± 4.9%. These figures indicate that the study students are carrying lighter schoolbags compared to corresponding students in other Saudi studies [3,6]. Additionally, only 37.1% of the study students were carrying more than the weight-limit recommendation for schoolbag use in children (10% of their body weight) [34]. This result was less than figures reported in Jeddah (75.9%) [6] and Al-Ahsa, Saudi Arabia (72.4%) [3].

There is controversy regarding the contributory role of relative schoolbag weight to back pain. It has been described as a risk factor for back pain by some studies [35,36]. However, more recent studies suggest no association between schoolbag weight and back pain or discomfort [11,37]. The present study found no evidence to support the role of heavy schoolbags (>10% of body weight) in developing back pain. This finding may suggest that the standard weight limit should be reviewed. The current study attempted to review the safe threshold limit of schoolbag weight and its discriminative power to identify back pain by conducting ROC curve analysis. According to the results of the ROC curve analysis, it was apparent that the ability of the relative schoolbag weight to predict back pain at different sites is poor for different cutoff points. These results are similar to the finding of a previous study [38], which reported that relative schoolbag weight alone is not enough as a guideline for safe schoolbag weight. Other factors should be taken into consideration along with relative weight, including carrying duration and method [38,39,40]. In addition, other studies suggest that schoolbag weight limits should take student age [41] and sex [42] into consideration.

## 5. Conclusions

Back pain is a prevalent problem among school-going adolescents in southwest Saudi Arabia. Most students carried their schoolbags on one side. One-sided carriage is a significant risk factor for middle back pain. Parents and teachers should encourage students to consider the correct schoolbag carriage. There is not sufficient evidence that the relative schoolbag weight alone is a predictive factor for back pain among school-going adolescents. Therefore, it is recommended that upcoming studies should consider the multifactorial etiology of back pain in adolescents. Future research should also take into account the person who packed the schoolbag (the child or the mother), whether it is packed for one day or the whole week, and the method of transportation to school (public transport, walking, biking, or parents’ car). Guidelines for safe schoolbags should be taken into consideration along with relative schoolbag weight and other factors such as student’s age and sex, as well as carrying duration and method.

## Figures and Tables

**Table 1 ijerph-17-00005-t001:** Prevalence of back pain among the study sample of school adolescents (n = 876).

Back Pain	No.	Prevalence % (95% CI)
Overall back pain	345	39.4 (35.4–43.7)
Upper back pain	127	14.5 (12.1–17.2)
Middle back pain	117	13.4 (11.1–15.5)
Lower back pain	101	11.5 (9.4–13.9)

**Table 2 ijerph-17-00005-t002:** Result of receiver operating characteristic (ROC) for schoolbag weight/student body weight percent to identify back pain.

Back Pain	AUC (95% CI)	Optimal Cutoff Point of Bag wt./Body wt. %	Sensitivity (%)	Specificity (%)
Overall	0.501 (0.467–0.535)	8.65%	53.62	53.62
Upper back pain	0.570 (0.5–0.603)	8.50%	60.63	51.50
Middle back pain	0.521 (0.4–0.554)	10.0%	70.9	37.55
Lower back pain	0.564 (0.5–0.597)	8.43%	62.38	54.19

AUC: Area under the curve; 95% CI: 95% confidence interval.

**Table 3 ijerph-17-00005-t003:** Characteristics of students’ schoolbags.

	Level		Total	*p*-Value
	Intermediate	Secondary		
Schoolbag’s full weight (kg)	4.95 ± 1.41	5.00 ± 1.77	4.97 ± 1.60	0.623
Schoolbag (kg)/body weight (kg) %:				0.001
<10	252 (57.4%)	299 (68.4%)	551 (62.9%)	
≥10	187 (42.6%)	138 (31.6%)	325 (37.1%)	
Schoolbag carriage method:				0.001
on back	254 (57.9%)	95 (21.7%)	349 (39.8%)	
on one side	185 (42.1%)	342 (78.3%)	527 (60.2%)	

**Table 4 ijerph-17-00005-t004:** Multivariable analysis for factors associated with back pain among the study sample of school adolescents (n = 876).

	Overall Back Pain	Upper Back Pain	Middle Back Pain	Lower Back Pain
	aOR (95% CI)	aOR (95% CI)	aOR (95% CI)	aOR (95% CI)
Sex: males	Ref	Ref	Ref	Ref
females	0.87 (0.60–1.27)	1.32 (0.79–2.20)	0.91 (0.53–1.54)	0.57 (0.32–1.04)
Level: secondary intermediate	Ref	Ref	Ref	Ref
	1.08 (0.80–1.46)	0.73 (0.48–1.12)	1.56 (1.04–2.40)	1.05 (0.66–1.67)
Method of carriage:				
on the back	Ref	Ref	Ref	Ref
on one side	1.20 (0.82–1.74)	0.81 (0.49–1.35)	2.13 (1.20–3.73)	0.89 (0.49–1.61)
Schoolbag (kg)/body				
weight (kg)%:				
<10	Ref	Ref	Ref	Ref
≥10	1.00 (0.75–1.40)	1.45 (0.98–2.14)	0.81 (0.53–1.25)	0.77 (0.50–1.23)

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
