# Peer review of "Back Pain and Schoolbags among Adolescents in Abha City, Southwestern Saudi Arabia"

_ijerph, 2019, doi:10.3390/ijerph17010005_

Round 1

Reviewer 1 Report

In this study, the authors studied the correlation between back pain and carrying schoolbags among adolescents aged between 13 to 19 years old. This study overlaps with the previous work from the same team in the International Journal of Medical Research & Health Sciences, titled as "School Screening for Scoliosis among Male Adolescents in Abha City, Southwestern Saudi Arabia". The research objectives and results of the submitted manuscript overlap with the previous work. The submitted manuscript lacks novelty and independence from the literature. The authors seem to were reluctant to cite their previous published work in this manuscript. 

I suggest the authors submit an addendum to the International Journal of Medical Research & Health Sciences. 

Author Response

Response to Reviewer 1 Comments

This study overlaps with the previous work from the same team in the International Journal of Medical Research & Health Sciences, titled as "School Screening for Scoliosis among Male Adolescents in Abha City, Southwestern Saudi Arabia". The research objectives and results of the submitted manuscript overlap with the previous work.

 AUTHORS’ RESPONSE

The current submitted work has no intended overlap with the previous work regrading objectives and results.

The objectives of the previous work were to study the prevalence of scoliosis and its associated factors among male adolescent in Abha city schools, while the objectives of the current study were to assess the problem of back pain among school adolescents, to find a new schoolbags weight/body mass optimal cutoff point to indicate back pain, to identify the methods of carrying schoolbags and to correlate back pain with these factors. The results of the previous work were focused on the prevalence of scoliosis among male school adolescents and its relations with male students’ grade, schoolbags, obesity and practicing physical activity, while the current study gives highlight on the problem of back pain among both male and female students with special emphasis on the role of schoolbags weight and carriage methods in pain development. The reason why the two studies seem to be overlapping is the study population in the previous work (male school adolescents) was a subcategory of the study population in the current one (both male and female school adolescents). The submitted manuscript lacks novelty and independence from the literature.

AUTHORS’ RESPONSE

The submitted manuscript helps in covering the knowledge gap regarding the magnitude of the back pain problem among school adolescents in southwest Saudi Arabia. It also supports the hypothesis of the multifactorial etiology of back pain. Additionally, the study recommended the upcoming guidelines for safe schoolbags should take into consideration along with relative schoolbags weight other factors like students’ age and sex, carrying duration, and method.

The authors seem to were reluctant to cite their previous published work in this manuscript.

AUTHORS’ RESPONSE

The authors used to cite the published works which are relevant to the discussed issues.  

I suggest the authors submit an addendum to the International Journal of Medical Research & Health Sciences. 

AUTHORS’ RESPONSE

Noted

Reviewer 2 Report

it is an interesting paper but I have a few recommendations to make it more attractive for the readers :

1

make an illustration of the human-product-interacton; in this case an (outline) drawing(so ethical more acceptable)  or picture of a series of children with a school bag to show the variation which there always is  and which is interesting for the readers. For example also including a light bag and a heavy bag, small kid and large kid. That are variations the reader would expect.

this drawing or picture is essential for the readers because each country/population has its own culture of typical  interacting with things depending on the place on the globe.  see for example on http://www.amydaneadventures.com/exhibit.php?id=6

2

the ROC graphs don't contribute much after reading the text and tables . I would recommend to remove them.

3

for your next research in this topic I would also recommend to observe or ask: - who is packing the schoolbag? the child or the mother? and is it packed for 1 day or for the whole week to avoid missing an item

what way of transport is used from home to school? public transport, walking, biking , by car of parents ?

If your like these questions put hem in your recommendations for future research at the end of the paper

Author Response

Response to Reviewer 2 Comments

Make an illustration of the human-product-interacton; in this case an (outline) drawing (so ethical more acceptable) or picture of a series of children with a school bag to show the variation which there always is  and which is interesting for the readers. For example, also including a light bag and a heavy bag, small kid and large kid. That are variations the reader would expect. this drawing or picture is essential for the readers because each country/population has its own culture of typical interacting with things depending on the place on the globe.  see for example on http://www.amydaneadventures.com/exhibit.php?id=6

AUTHORS’ RESPONSE

Regarding the issue of including children pictures, for ethical and cultural reasons, especially with no previous consent taken from students and their parents, authors feel that this issue will raise potential problems.

Although the idea of drawing the students' school bag is interesting for the reader, including these drawings in the manuscript is difficult as it necessitates a professional work and experience that might exceed the capacity of the research team.

The ROC graphs don't contribute much after reading the text and tables . I would recommend to remove them.

AUTHORS’ RESPONSE

We appreciate your recommendation. The ROC graphs were removed (pages 9 and 10)

For your next research in this topic I would also recommend to observe or ask: - who is packing the schoolbag? the child or the mother? and is it packed for 1 day or for the whole week to avoid missing an item- what way of transport is used from home to school? public transport, walking, biking, by car of parents ? If you’re like these questions put them in your recommendations for future research at the end of the paper.

AUTHORS’ RESPONSE

The authors appreciate the recommended questions and observation to be considered in the future research. The following passage was added in the recommendation “Also, future researches should take into account the person who packed the schoolbag (the child or the mother), and is it packed for 1 day or the whole week? and the method of transportation to school (public transport, walking, biking, parents' car).” Page 5, lines 217-220

Reviewer 3 Report

Thank you for the opportunity to review this paper. My comments are as follows:

Abstract: Line 22 probably should be "To assess the problem of BP and associated factors..."

Introduction: Appropriate.

Materials and Methods:

Line 67, "minimal" should be "minimum"

Line 70, male female should be male/female.

Line 102, How are intermediate or secondary defined? What year level/age are these two groups?

There needs to be greater description of the methodology in general. How many students were involved? It is stated that minimum sample size was 600 - was that the final sample number? What ages? What year levels? How many schools/districts? Did trained examiners measure students. Was the questionnaire paper-based in a class room?

Results:

Line 130, "statically" should be "statistically"

Line 136 should be "most of the intermediate.."

Discussion: Line 194 "%" sign repeated

Conclusion: Appropriate.

Tables and figures appropriate.

Some minor issues with English.

Author Response

Response to Reviewer 3 Comments

Abstract: Line 22 probably should be "To assess the problem of BP and associated factors..."

AUTHORS’ RESPONSE

This was taken into consideration. Line 22 changed to read “To assess the problem of BP and associated factors among adolescents”. (page 1, lines 22-19)

Line 67, "minimal" should be "minimum"

AUTHORS’ RESPONSE

Corrected

Line 70, male female should be male/female.

AUTHORS’ RESPONSE

Corrected

Line 102, How are intermediate or secondary defined? What year level/age are these two groups?

AUTHORS’ RESPONSE

In line 102 the word “grade” should read “level”. Accordingly, the similar changes were done in the text and tables. The following passage was added to the methods section to define the groups. “The intermediate and secondary levels in Saudi Arabia include students in grades from 7 to 9 and 10 to 12, respectively. The students’ age ranges from 13–15 years at the intermediate level and 16–18 years at the secondary level.” (page 2, lines 64-66)

There needs to be greater description of the methodology in general. How many students were involved? It is stated that minimum sample size was 600 - was that the final sample number? What ages? What year levels? How many schools/districts? Did trained examiners measure students. Was the questionnaire paper-based in a class room?

AUTHORS’ RESPONSE

Regarding the final sample number and their ages, as shown in the first paragraph of result section “The present study included 876 adolescents. They were students in intermediate level (439, 50.1%) and secondary schools (437, 49.9%). The sample comprised 471 males (53.8%) and 405 females (46.2%). Their age ranged from 13 to 19 years with a mean of 15.88 ± 1.76 years and a median of 16 years. (lines 109-113) Regarding the question “how many schools/districts. The following passage was added to the methods section “The students were recruited from 13 intermediate and secondary schools in Abha city. Within each school, one class was selected randomly from each of the three grades.” Lines 74-76 Regarding the question “Did trained examiners measure students”, the following statement was added in line 88 “All measurements were taken by trained examiners”. Regarding the question “was questionnaire paper-based in a class room? This w as clarified in lines 78 and 79. “An anonymous paper-based questionnaire was distributed for each student in his own classroom.” Line 130, "statically" should be "statistically"

AUTHORS’ RESPONSE

 Corrected

Line 136 should be "most of the intermediate.."

AUTHORS’ RESPONSE

 Corrected

Discussion: Line 194 "%" sign repeated

AUTHORS’ RESPONSE

Corrected

Some minor issues with English.

AUTHORS’ RESPONSE

Revised

Round 2

Reviewer 1 Report

As the authors stated in their rebuttal letter, the previously published work was focused on scoliosis prevalence and its relations with male students’ grade, schoolbags, obesity and practicing physical activity.

However, it does not mean that the significance level of findings in the original study is enough to conduct data post-processing on a portion of the results and publish it as a new manuscript. 

With that logic next paper would be the relation between scoliosis and participating in physical activity in Abha city, and so on. I think this study is a multi-variable study and all factors should be included in the statistical analyses and discussions. 

Overall, the weak study objectives would cause low interest to the readers and arise questions regarding the validity of conclusions. 

Reviewer 3 Report

I thank the authors for addressing the previous comments. My remaining comments are as follows:

Abstract: Appropriate

Introduction: Line 44 "revealed" should be "reveal"

Line 48 should be "A bag carried on one side is associated..."

Line 50 should be "..adolescents is an important.."

Some minor issues with English.

Methods: Line 74 requires a bit more clarification ".. each of the three grades (7-9 and 10-12).

Minor issues with English.

Results: Appropriately presented.

Discussion: Line 190 "back" should be "two shoulders" Line 191 and 193 "pains" should be "pain"

Line 197-200 Sentence not clear, and needs to be reworked.

Line 201 "..is existing.." should be "exists"

Line 205 "..the standard of weight limit ..." should be "..the standard weight limit.."

Line 206-207 "..to back pain.." should be "to identify back pain.."

Minor issues with English

Conclusion: Line 221 "researches" should be "research"

Line 222 ? not required in the sentence
